# Deep neuroethology of a virtual rodent

**Josh Merel**[*,1], **Diego Aldarondo**[*,2,3], **Jesse Marshall**[*,3,4], **Yuval Tassa**[1],
**Greg Wayne**[1], **Bence Ölveczky**[3,4]
[1]DeepMind, London, UK.
[2]Program in Neuroscience, [3]Center for Brain Science, [4]Department of Organismic and
Evolutionary Biology, Harvard University, Cambridge, MA 02138, USA.
`jsmerel@google.com, diegoaldarondo@g.harvard.edu,`
`jesse_d_marshall@fas.harvard.edu`

## Abstract

Parallel developments in neuroscience and deep learning have led to mutually productive exchanges, pushing our understanding of real and artificial neural networks in sensory and cognitive systems. However, this interaction between fields is less developed in the study of motor control. In this work, we develop a virtual rodent as a platform for the grounded study of motor activity in artificial models of embodied control. We then use this platform to study motor activity across contexts by training a model to solve four complex tasks. Using methods familiar to neuroscientists, we describe the behavioral representations and algorithms employed by different layers of the network using a neuroethological approach to characterize motor activity relative to the rodent's behavior and goals. We find that the model uses two classes of representations which respectively encode the task-specific behavioral strategies and task-invariant behavioral kinematics. These representations are reflected in the sequential activity and population dynamics of neural subpopulations. Overall, the virtual rodent facilitates grounded collaborations between deep reinforcement learning and motor neuroscience.

## 1 Introduction

Animals have nervous systems that allow them to coordinate their movement and perform a diverse set of complex behaviors. Mammals, in particular, are generalists in that they use the same general neural network to solve a wide variety of tasks. This flexibility in adapting behaviors towards many different goals far surpasses that of robots or artificial motor control systems. Hence, studies of the neural underpinnings of flexible behavior in mammals could yield important insights into the classes of algorithms capable of complex control across contexts and inspire algorithms for flexible control in artificial systems (Merel et al., 2019b).

Recent efforts at the interface of neuroscience and machine learning have sparked renewed interest in constructive approaches in which artificial models that solve tasks similar to those solved by animals serve as normative models of biological intelligence. Researchers have attempted to leverage these models to gain insights into the functional transformations implemented by neurobiological circuits, prominently in vision (Khaligh-Razavi & Kriegeskorte, 2014; Yamins et al., 2014; Kar et al., 2019), but also increasingly in other areas, including audition (Kell et al., 2018) and navigation (Banino et al., 2018; Cueva & Wei, 2018). Efforts to construct models of biological locomotion systems have informed our understanding of the mechanisms and evolutionary history of bodies and behavior (Grillner et al., 2007; Ijspeert et al., 2007; Ramdya et al., 2017; Nyakatura et al., 2019). Neural control approaches have also been applied to the study of reaching movements, though often in constrained behavioral paradigms (Lillicrap & Scott, 2013), where supervised training is possible (Sussillo et al., 2015; Michaels et al., 2019).

While these approaches model parts of the interactions between animals and their environments (Chiel & Beer, 1997), none attempt to capture the full complexity of embodied control, involving how an animal uses its senses, body and behaviors to solve challenges in a physical environment.

---

[*]Equal contribution.

The development of models of embodied control is valuable to the field of motor neuroscience, which typically focuses on restricted behaviors in controlled experimental settings. It is also valuable for AI research, where flexible models of embodied control could be applicable to robotics.

Here, we introduce a virtual model of a rodent to facilitate grounded investigation of embodied motor systems. The virtual rodent affords a new opportunity to directly compare principles of artificial control to biological data from real-world rodents, which are more experimentally accessible than humans. We draw inspiration from emerging deep reinforcement learning algorithms which now allow artificial agents to perform complex and adaptive movement in physical environments with sensory information that is increasingly similar to that available to animals (Peng et al., 2016; 2017; Heess et al., 2017; Merel et al., 2019a;c). Similarly, our virtual rodent exists in a physical world, equipped with a set of actuators that must be coordinated for it to behave effectively. It also possesses a sensory system that allows it to use visual input from an egocentric camera located on its head and proprioceptive input to sense the configuration of its body in space.

There are several questions one could answer using the virtual rodent platform. Here we focus on the problem of embodied control across multiple tasks. While some efforts have been made to analyze neural activity in reduced systems trained to solve multiple tasks (Song et al., 2017; Yang et al., 2019), those studies lacked the important element of motor control in a physical environment. Our rodent platform presents the opportunity to study how representations of movements as well as sequences of movements change as a function of goals and task contexts.

To address these questions, we trained our virtual rodent to solve four complex tasks within a physical environment, all requiring the coordinated control of its body. We then ask "Can a neuroscientist understand a virtual rodent?" – a more grounded take on the originally satirical "Can a biologist fix a radio?" (Lazebnik, 2002) or the more recent "Could a neuroscientist understand a microprocessor?" (Jonas & Kording, 2017). We take a more sanguine view of the tremendous advances that have been made in computational neuroscience in the past decade, and posit that the supposed 'failure' of these approaches in synthetic systems is partly a misdirection. Analysis approaches in neuroscience were developed with the explicit purpose of understanding sensation and action in real brains, and often implicitly rooted in the types of architectures and processing that are thought relevant in biological control systems. With this philosophy, we use analysis approaches common in neuroscience to explore the types of representations and dynamics that the virtual rodent's neural network employs to coordinate multiple complex movements in the service of solving motor and cognitive tasks.

## 2 APPROACH

### 2.1 VIRTUAL RODENT BODY

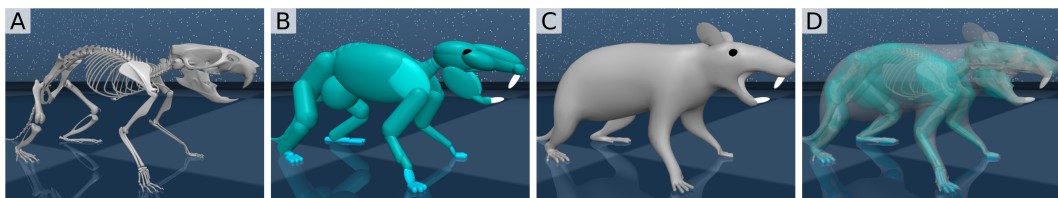

Figure 1: (A) Anatomical skeleton of a rodent (as reference; not part of physical simulation). (B) A body designed around the skeleton to match the anatomy and model collisions with the environment. (C) Purely cosmetic skin to cover the body. (D) Semi-transparent visualization of (A)-(C) overlain.

We implemented a virtual rodent body (Figure 1) in MuJoCo (Todorov et al., 2012), based on measurements of laboratory rats (see Appendix A.1). The rodent body has 38 controllable degrees of freedom. The tail, spine, and neck consist of multiple segments with joints, but are controlled by tendons that co-activate multiple joints (spatial tendons in MuJoCo). The rodent will be released as part of `dm_control/locomotion`.

The virtual rodent has access to proprioceptive information as well as "raw" egocentric RGB-camera (64×64 pixels) input from a head-mounted camera. The proprioceptive inputs include internal

joint angles and angular velocities, the positions and velocities of the tendons that provide actuation, egocentric vectors from the root (pelvis) of the body to the positions of the head and paws, a vestibular-like upright orientation vector, touch or contact sensors in the paws, as well as egocentric acceleration, velocity, and 3D angular velocity of the root.

## 2.2 VIRTUAL RODENT TASKS

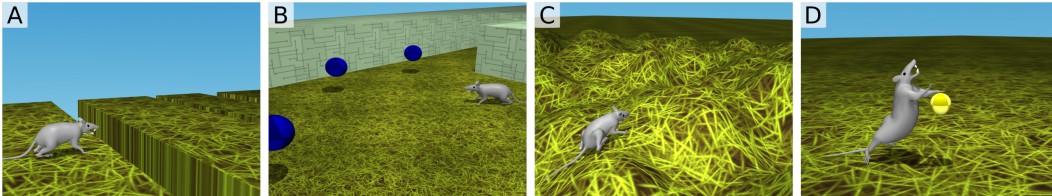

Figure 2: Visualizations of four tasks the virtual rodent was trained to solve: (A) jumping over gaps ("gaps run"), (B) foraging in a maze ("maze forage"), (C) escaping from a hilly region ("bowl escape"), and (D) touching a ball twice with a forepaw with a precise timing interval between touches ("two-tap").

We implemented four tasks adapted from previous work in deep reinforcement learning and motor neuroscience (Merel et al., 2019a; Tassa et al., 2018; Kawai et al., 2015) to encourage diverse motor behaviors in the rodent. The tasks are as follows: (1) Run along a corridor, over "gaps", with a reward for traveling along the corridor at a target velocity (Figure 2A). (2) Collect all the blue orbs in a maze, with a sparse reward for each orb collected (Figure 2B). (3) Escape a bowl-shaped region by traversing hilly terrain, with a reward proportional to distance from the center of the bowl (Figure 2C). (4) Approach orbs in an open field, activate them by touching them with a forepaw, and touch them a second time after a precise interval of 800ms with a tolerance of $\pm$100ms; there is a time-out period if the touch is not within the tolerated window and rewards are provided sparsely on the first and second touch (Figure 2D). We did not provide the agent with a cue or context indicating its task. Rather, the agent had to infer the task from the visual input and behave appropriately.

## 2.3 TRAINING A MULTI-TASK POLICY

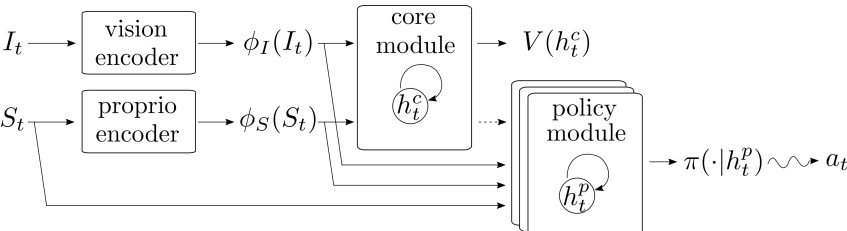

Figure 3: The virtual rodent agent architecture. Egocentric visual image inputs are encoded into features via a small residual network (He et al., 2016) and proprioceptive state observations are encoded via a small multi-layer perceptron. The features are passed into a recurrent LSTM module (Hochreiter & Schmidhuber, 1997). The core module is trained by backpropagation during training of the value function. The outputs of the core are also passed as features to the policy module (with the dashed arrow indicating no backpropogation along this path during training) along with shortcut paths from the proprioceptive observations as well as encoded features. The policy module consists of one or more stacked LSTMs (with or without skip connections) which then produce the actions via a stochastic policy.

Emboldened by recent results in which end-to-end RL produces a single terrain-adaptive policy (Peng et al., 2016; 2017; Heess et al., 2017), we trained a single architecture on the multiple motor-control-reliant tasks (see Figure 3). To train a single policy to perform all four tasks, we used an IMPALA-style setup for actor-critic DeepRL (Espeholt et al., 2018); parallel workers collected

rollouts, logged them to a replay, from which a central learner sampled data to perform updates. The value-function critic was trained using off-policy correction via V-trace. To update the actor, we used a variant of MPO (Abdolmaleki et al., 2018) where the E-step is performed using advantages determined from the empirical returns and the value-function, instead of the Q-function (Song et al., 2019). Empirically, we found that the "escape" task was more challenging to learn during interleaved training relative to the other tasks. Consequently, we present results arising from training a single-task expert on the escape task and training the multi-task policies using kickstarting for that task (Schmitt et al., 2018), with a weak coefficient (.001 or .005). Kickstarting on this task made the seeds more reliably solve all four tasks, facilitating comparison of the multi-task policies with different architectures (i.e. the policy having 1, 2, or 3 layers, with or without skip connections across those layers). The procedure yields a single neural network that uses visual inputs to determine how to behave and coordinates its body to move in ways required to solve the tasks. See video examples of a single policy solving episodes of each task: gaps, forage, escape, and two-tap.

## 3 ANALYSIS

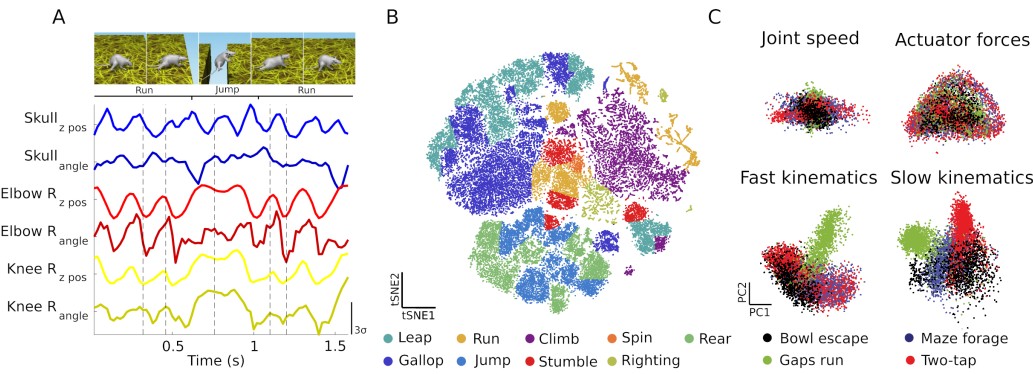

Figure 4: Ethology of the virtual rodent. (A) Example jumping sequence in gaps run task with a representative subset of recorded behavioral features. Dashed lines denote the time of the corresponding frames (top). (B) tSNE embedding of 60 behavioral features describing the pose and kinematics of the virtual rodent allows identification of rodent behaviors. Points are colored by hand-labeling of behavioral clusters identified by watershed clustering. (C) The first two principal components of different behavioral features reveals that behaviors are more shared across tasks at short, 5-25 Hz timescales (fast kinematics), but no longer 0.3-5 Hz timescales (slow kinematics).

We analyzed the virtual rodent's neural network activity in conjunction with its behavior to characterize how it solves multiple tasks (Figure 4A). We used analyses and perturbation techniques adapted from neuroscience, where a range of techniques have been developed to highlight the properties of real neural networks. Biological neural networks have been hypothesized to control, select, and modulate movement through a variety of debated mechanisms, ranging from explicit neural representations of muscle forces and behavioral primitives, to more abstract production of neural dynamics that could underly movement (Graziano, 2006; Kalaska, 2009; Churchland et al., 2012). A challenge with nearly all of these models however is that they have largely been inspired by findings from individual behavioral tasks, making it unclear how to generalize them to a broader range of naturalistic behaviors. To provide insight into mechanisms underlying movement in the virtual rodent, and to potentially give insight by proxy into the mechanisms underlying behavior in real rats, we thus systematically tested how the different network layers encoded and generated different aspects of movement.

For all analyses we logged the virtual rodent's kinematics, joint angles, computed forces, sensory inputs, and the cell unit activity of the LSTMs in core and policy layers during 25 trials per task from each network architecture.

### 3.1 Virtual rodents exhibit behavioral flexibility.

We began our analysis by quantitatively describing the behavioral repertoire of the virtual rodent. A challenge in understanding the neural mechanisms underlying behavior is that it can be described at many timescales. On short timescales, one could describe rodent locomotion using a set of actuators that produce joint-specific patterns of forces and kinematics. However on longer timescales, these force patterns are organized into coordinated, re-used movements, such as running, jumping, and turning. These movements can be further combined to form behavioral strategies or goal-directed behaviors. Relating neural representations to motor behaviors therefore requires analysis methods that span multiple timescales of behavioral description. To systematically examine the classes of behaviors these networks learn to generate and how they are differentially deployed across tasks, we developed sets of behavioral features that describe the kinematics of the animal on fast (5-25 Hz), intermediate (1-25 Hz) or slow (0.3-5 Hz) timescales (Appendix A.2, A.3 ). As validation that these features reflected meaningful differences across behaviors, embedding these features using tSNE (Maaten & Hinton, 2008) produced a behavioral map in which virtual rodent behaviors, were segregated to different regions of the map (Figure 4B)(see video). This behavioral repertoire of the virtual rodent consisted of many behaviors observed in rodents, such as rearing, jumping, running, climbing and spinning. While the exact kinematics of the virtual rodent's behaviors did not exactly match those observed in real rats, they did reproduce unexpected features. For instance the stride frequency of the virtual rodent during galloping matches that observed in rats (Appendix A.3).

We next investigated how these behaviors were used by the virtual rodent across tasks. On short timescales, low-level motor features like joint speed and actuator forces occupied similar regions in principal component space (Figure 4C). In contrast, behavioral kinematics, especially on long, 0.3-5 Hz timescales, were more differentiated across tasks. Similar results held when examining overlap in other dimensions using multidimensional scaling. Overall this suggests that the network learned to adapt similar movements in a selective manner for different tasks, suggesting that the agent exhibited a form of behavioral flexibility.

### 3.2 Networks primarily reflect behaviors, not forces

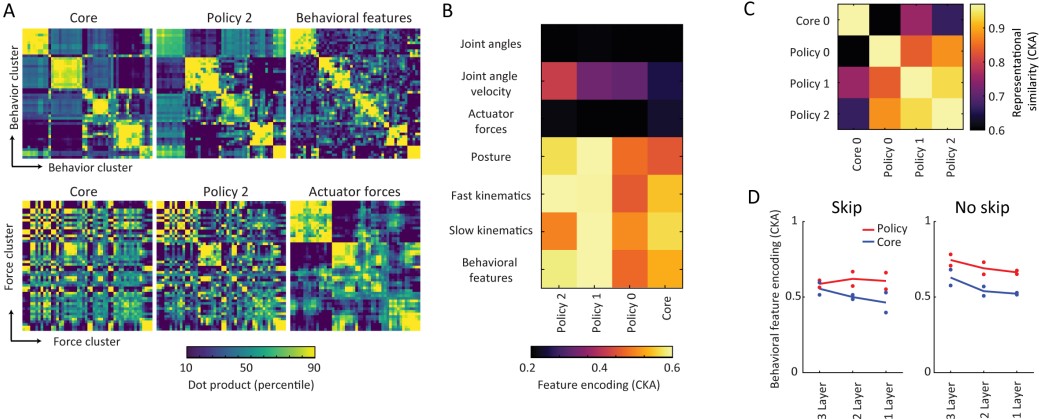

Figure 5: Representational structure of the rodent's neural network. (A) Example similarity matrices of neural networks and behavioral descriptors. We grouped behavioral descriptors into 50 clusters that and we computed the average neural population vector during each cluster (AppendixA.4). Similarity was assessed by computing the dot product of either the neural population vector or the behavioral feature vector within each cluster. (B) Centered Kernel Alignment (CKA) index of neural and behavioral feature similarity matrices for 3 and 1 policy layer architectures. (C) CKA index of feature similarity matrices across all pairs of network layers. (D) Average CKA index between core and policy layers and behavioral features, compared across architectures. Points show values from individual network seeds. Policy values are averaged across layers.

We next examined the neural activity patterns underlying the virtual rodent's behavior to test if networks produced behaviors through explicit representations of forces, kinematics or behaviors. As

expected, core and policy units operate on distinct timescales (See Appendix A.3, Figure 9). Units in the core typically fluctuated over timescales of 1-10 seconds, likely representing variables associated with context and reward. In contrast, units in policy layers were more active over subsecond timescales, potentially encoding motor and behavioral features.

To quantify which aspects of behavior were encoded in the core and policy layers, and how these patterns varied across layers, we used representational similarity analysis (RSA) (Kriegeskorte et al., 2008; Kriegeskorte & Diedrichsen, 2019). RSA provides a global measure of how well different features are encoded in layers of a neural network by analyzing the geometries of network activity upon exposure to several stimuli, such as objects. To apply RSA, first a representational similarity (or equivalently, dissimilarity) matrix is computed that quantifies the similarity of neural population responses to a set of stimuli. To test if different neural populations show similar stimulus encodings, these similarity matricies can then be directly compared across different network layers. Multiple metrics, such as the matrix correlation or dot product can be used to compare these neural representational similarity matricies. Here we used the linear centered kernel alignment (CKA) index, which shows invariance to orthonormal rotations of population activity (Kornblith et al., 2019).

RSA can also be used to directly test how well a particular stimulus feature is encoded in a population. If each stimuli can be quantitively described by one or more feature vectors, a similarity matrix can also be computed across the set of stimuli themselves. The strength of encoding of a particular set of features can by measured by comparing the correlation of the stimulus feature similarity matrix and the neuronal similarity matrix. The correlation strength directly reflects the ability of a linear decoder trained on the neuronal population vector to distinguish different stimuli (Kriegeskorte & Diedrichsen, 2019). Unlike previous applications of RSA in the analysis of discrete stimuli such as objects, (Khaligh-Razavi & Kriegeskorte, 2014; Yamins et al., 2014) behavior evolves continuously. To adapt RSA to behavioral analysis, we partitioned time by discretizing each behavioral feature into 50 clusters (Appendix A.4).

As expected, RSA revealed that core and policy layers encoded somewhat distinct behavioral features. Policy layers contained greater information about fast timescale kinematics in a manner that was largely conserved across layers, while core layers showed more moderate encoding of kinematics that was stronger for slow behavioral features (Figure 5B,C). This difference in encoding was largely consistent across all architectures tested (Figure 5D).

The feature encoding of policy networks was somewhat consistent with the emergence of a hierarchy of behavioral abstraction. In networks trained with three policy layers, representations were distributed in timescales across layers, with the last layer (policy 2) showing stronger encoding of fast behavioral features, and the first layer (policy 0) instead showing stronger encoding of slow behavioral features. However, policy layer activity, even close to the motor periphery, did not show strong explicit encoding of behavioral kinematics or forces.

## 3.3 BEHAVIORAL REPRESENTATIONS ARE SHARED ACROSS TASKS

We then investigated the degree to which the rodent's neural networks used the same neural representations to produce behaviors, such as running or spinning, that were shared across tasks. Embedding population activity into two-dimensions using multidimensional scaling revealed that core neuron representations were highly distinct across all tasks, while policy layers contained more overlap (Figure 6A), suggesting that some behavioral representations were re-used. Comparison of representational similarity matricies for behaviors that were shared across tasks revealed that policy layers tended to possess a relatively similar encoding of behavioral features, especially fast behavioral features, over tasks (Figure 6C; Appendix A.4). This was validated by inspection of neural activity during individual behaviors shared across tasks (Appendix A.5, Figure 10). Core layer representations across almost all behavioral categories were more variable across tasks, consistent with encoding behavioral sequences or task variables.

Interestingly, when comparing this cross-task encoding similarity across architectures, we found that one layer networks showed a marked increase in the similarity of behavioral encoding across tasks (Figure 6D). This suggests that in networks with lower computational capacity, animals must rely on a smaller, shared behavioral representation across tasks.

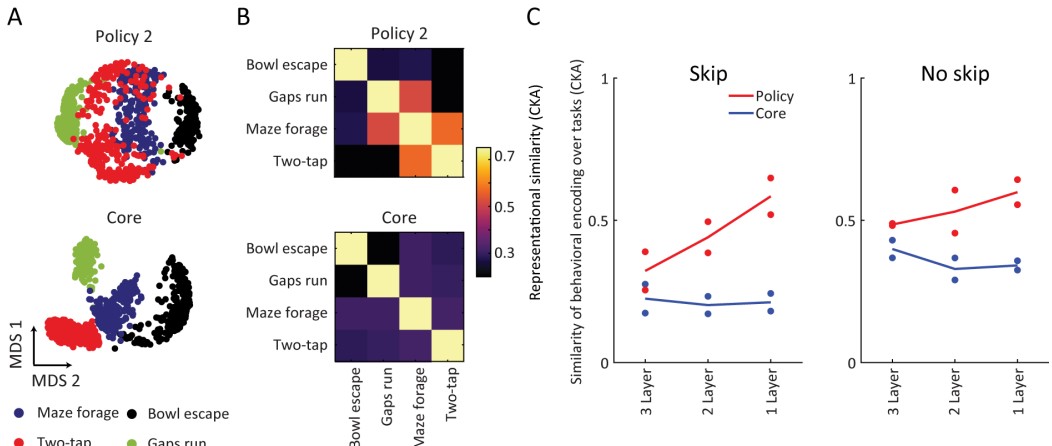

Figure 6: Policy representations are shared across tasks. (A) Two-dimensional multidimensional scaling embeddings of core and policy activity shows that while policy representations overlap across some tasks, core representations are largely distinct. (B) CKA index of the policy 2 and core network representations of behavioral features during behaviors shared across different tasks (Appendix A.4). Policy 2, but not core networks show similar encoding patterns across the across the maze forage and two-tap tasks, as well as the gaps run and maze forage tasks, consistent with the shared behaviors used across these tasks. (C) The similarity of behavioral feature encoding (CKA index) across different architectures demonstrates that networks with fewer layers show greater similarity across tasks. Points show values from individual seeds.

## 3.4 NEURAL POPULATION DYNAMICS ARE SYNCHRONIZED WITH BEHAVIOR

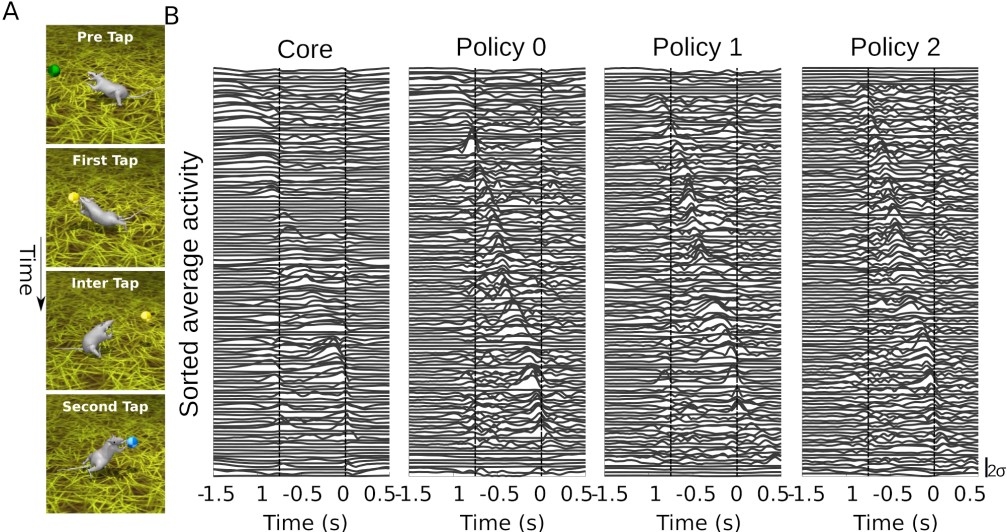

Figure 7: Neurons in core and policy networks show sequential activity during stereotyped behavior. (A) Example video stills showing the virtual rodent engaged in the two-tap task (B) Average absolute z-scored activity traces of all 128 neurons in each layer during performance of the two-tap sequence. Traces are sorted by the time of peak average firing rate. Dashed lines indicate the times of first and second taps. Sequential neural activity is present during the two-tap sequence.

While RSA described which behavioral features were represented in core and policy activity, we were also interested in describing how neural activity changes over time to produce different behaviors. We began by analyzing neural activity during the production of stereotyped behaviors. Activity

patterns in the two-tap task showed peak activity in core and policy units that was sequentially organized (Figure 7), uniformly tiling time between both taps of the two-tap sequence. This sequential activation was observed across tasks and behaviors in the policy network, including during running (see video) where, consistent with policy networks encoding short-timescale kinematic features in a task-invariant manner, neural activity sequences were largely conserved across tasks (See Appendix A.5, Figure 10). These sequences were reliably repeated across instances of the respective behaviors, and in the case of the two-tap sequence, showed reduced neural variability relative to surrounding timepoints (See Appendix A.6, Figure 11).

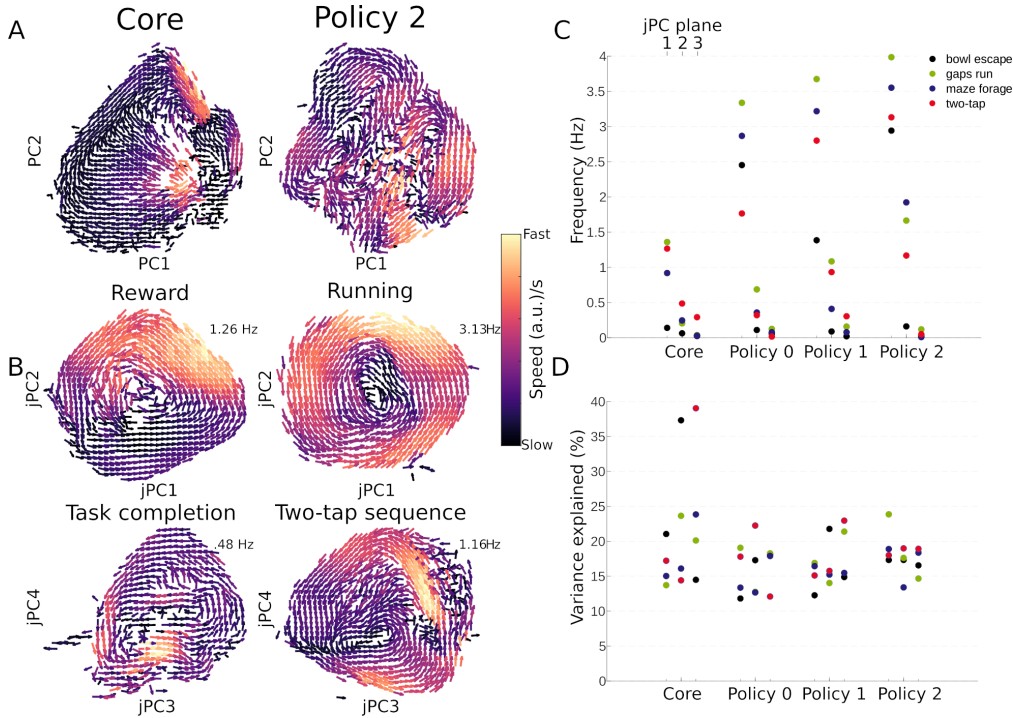

Figure 8: Latent network dynamics within tasks reflect rodent behavior on different timescales. (A) Vector field representation of the first two principal components of neural activity in the core and final policy layers during the two-tap task. PC spaces show signatures of rotational dynamics. (B) Vector field representation of first two jPC planes for the core and final policy layers during the two-tap task. Apparent rotations within the different planes are associated with behaviors and behavioral features of different timescales, labeled above. Columns denote layer (as in (A)), while rows denote jPC plane. (C) Characteristic frequency of rotations within each jPC plane. Groups of three points respectively indicate the first, second, and third jPC planes for a given layer. Rotations in the core are slower than those in the policy. (D) Variance explained by each jPC plane.

The finding of sequential activity hints at a putative mechanism for the rodent's behavioral production. We next hoped to systematically quantify the types of sequential and dynamical activity present in core and policy networks without presupposing the behaviors of interest. To describe population dynamics in relation to behavior, we first applied principal components analysis (PCA) to the activity during the performance of single tasks, and visualized the gradient of the population vector as a vector field. Figure 8A shows such a vector field representation of the first two principal components of the core and final policy layer during the two-tap task. We generated vector fields by discretizing the PC space into a two-dimensional grid and calculating the average neural activity gradient with respect to time for each bin.

The vector fields showed strong signatures of rotational dynamics across all layers, likely a signature of previously described sequential activity. To extract rotational patterns, we used jPCA, a dimensionality reduction method that extracts latent rotational dynamics in neural activity (Churchland et al., 2012). The resulting jPCs form an orthonormal basis that spans the same space as the first six traditional PCs, while maximally emphasizing rotational dynamics. Figure 8B shows the vector

fields of the first two jPC planes for the core and final policy layers along with their characteristic frequency. Consistent with our previous findings, jPC planes in the core have lower characteristic frequencies than those in policy layers across tasks (Figure 8C). The jPC planes also individually explained a large percentage of total neural variability (Figure 8D).

These rotational dynamics in the policy and core jPC planes were respectively associated with the production of behaviors and the reward structure of the task. For example, in the two-tap task, rotations in the fastest jPC plane in the core were concurrent with the approach to reward, while rotations in the second fastest jPC were concurrent with long timescale transitions between running to the orb and performing the two-tap sequence. Similarly, the fastest jPC in policy layers was correlated with the phase of running, while the second fastest was correlated with the phase of the two-tap sequence (video). This trend of core and policy neural dynamics respectively reflecting behavioral and task-related features was also present in other tasks. For example, in the maze forage task, the first two jPC planes in the core respectively correlated with reaching the target orb and discovering the location of new orbs, while those in the policy were correlated with low-level locomotor features such as running phase (video). Along with RSA, these findings support a model in which the core layer transforms sensory information into a contextual signal in a task-specific manner. This signal then modulates activity in the policy toward different trajectories that generate appropriate behaviors in a more task-independent fashion. For a more complete set of behaviors with neural dynamics visualizations overlaid, see Appendix A.7.

### 3.5 NEURAL PERTURBATIONS CORROBORATE DISTINCT ROLES ACROSS LAYERS

To causally demonstrate the differing roles of core and policy units in respectively encoding task-relevant features and movement, we performed silencing and activation of different neuronal subsets in the two-tap task. We identified two stereotyped behaviors (rears and spinning jumps) that were reliably used in two different seeds of the agent to reach the orb in the task. We ranked neurons according to the degree of modulation of their z-scored activity during the performance of these behaviors. We then inactivated subsets of neurons by clamping activity to the mean values between the first and second taps and observed the effects of inactivation on trial success and behavior.

In both seeds analyzed, inactivation of policy units had a stronger effect on motor behavior than the inactivation of core units. For instance, in the two-tap task, ablation of 64 neurons in the final policy layer disrupts the performance of the spinning jump (Appendix A.8 Figure 12B video). In contrast, ablation of behavior-modulated core units did not prevent the production of the behavior, but mildly affected the way in which the behavior is directed toward objects in the environment. For example, ablation of a subset of core units during the performance of a spinning jump had a limited effect, but sometimes resulted in jumps that missed the target orbs (video; See Appendix A.8, Figure 12C).

We also performed a complementary perturbation aimed to elicit behaviors by overwriting the cell state of neurons in each layer with the average time-varying trajectory of neural activity measured during natural performance of a target behavior. The efficacy of stimulation was found to depend on the gross body posture and behavioral state of an animal, but was nevertheless successful in some cases. For example, during the two-tap sequence, we were able to elicit spinning movements common to searching behaviors in the forage task (video; See Appendix A.8, Figure 12D, E). The efficacy of this activation was more reliable in layers closer to the motor output (Figure 12D). In fact, activation of core units rarely elicited spins, but rather elicited sporadic dashes reminiscent of the searching strategy of many models during the forage task (video).

## 4 DISCUSSION

For many computational neuroscientists and artificial intelligence researchers, an aim is to reverse-engineer the nervous system at an appropriate level of abstraction. In the motor system, such an effort requires that we build embodied models of animals equipped with artificial nervous systems capable of controlling their synthetic bodies across a range of behavior. Here we introduced a virtual rodent capable of performing a variety of complex locomotor behaviors to solve multiple tasks using a single policy. We then used this virtual nervous system to study principles of the neural control of movement across contexts and described several commonalities between the neural activity of artificial control and previous descriptions of biological control.

A key advantage of this approach relative to experimental approaches in neuroscience is that we can fully observe sensory inputs, neural activity, and behavior, facilitating more comprehensive testing of theories related to how behavior can be generated. Furthermore, we have complete knowledge of the connectivity, sources of variance, and training objectives of each component of the model, providing a rare ground truth to test the validity of our neural analyses. With these advantages in mind, we evaluated our analyses based on their capacity to both describe the algorithms and representations employed by the virtual rodent and recapitulate the known functional objectives underlying its creation without prior knowledge.

To this end, our description of core and policy as respectively representing value and motor production is consistent with the model's actor-critic training objectives. But beyond validation, our analyses provide several insights into how these objectives are reached. RSA revealed that the cell activity of core and policy layers had greater similarity with behavioral and postural features than with short-timescale actuators. This suggests that the representation of behavior is useful in the moment-to-moment production of motor actions in artificial control, a model that has been previously proposed in biological action selection and motor control (Mink, 1996; Graziano, 2006). These behavioral representations were more consistent across tasks in the policy than in the core, suggesting that task context and value activity in the core engaged task-specific behavioral strategies through the reuse of shared motor activity in the policy.

Our analysis of neural dynamics suggests that reused motor activity patterns are often organized as sequences. Specifically, the activity of policy units uniformly tiles time in the production of several stereotyped behaviors like running, jumping, spinning, and the two-tap sequence. This finding is consistent with reports linking sequential neural activity to the production of stereotyped motor and task-oriented behavior in rodents (Berke et al., 2009; Rueda-Orozco & Robbe, 2015; Dhawale et al., 2019), including during task delay periods (Akhlaghpour et al., 2016), as well as in singing birds (Albert & Margoliash, 1996; Hahnloser et al., 2002). Similarly, by relating rotational dynamics to the virtual rodent's behavior, we found that different behaviors were seemingly associated with distinct rotations in neural activity space that evolved at different timescales. These findings are consistent with a hierarchical control scheme in which policy layer dynamics that generate reused behaviors are activated and modulated by sensorimotor signals from the core.

This work represents an early step toward the constructive modeling of embodied control for the purpose of understanding the neural mechanisms behind the generation of behavior. Incrementally and judiciously increasing the realism of the model's embodiment, behavioral repertoire, and neural architecture is a natural path for future research. Our virtual rodent possesses far fewer actuators and touch sensors than a real rodent, uses a vastly different sense of vision, and lacks integration with olfactory, auditory, and whisker-based sensation (see Zhuang et al., 2017). While the virtual rodent is capable of locomotor behaviors, an increased diversity of tasks involving decision making, memory-based navigation, and working memory could give insight into "cognitive" behaviors of which rodents are capable. Furthermore, biologically-inspired design of neural architectures and training procedures should facilitate comparisons to real neural recordings and manipulations. We expect that this comparison will help isolate residual elements of animal behavior generation that are poorly captured by current models of motor control, and encourage the development of artificial neural architectures that can produce increasingly realistic behavior.

## AUTHOR CONTRIBUTIONS

Josh and Yuval built the rodent MuJoCo model, with measurements collected by Diego and Jesse. Josh trained the virtual rodent model. Jesse performed behavioral and neural representation analyses. Diego performed neural dynamics analyses. Josh, Jesse, and Diego drafted the manuscript. All authors contributed to the conception of the project.

## ACKNOWLEDGMENTS

The rodent skeleton reference model was purchased from leo3Dmodels on TurboSquid. Thanks to Max Cant for the rodent skin, and Marcus Wainwright for the skybox and ground textures. D.A. was supported by NSF GRFP DGE1745303. J.D.M was supported by a fellowship from the Helen Hay Whitney foundation sponsored by Vertex and a K99/R00 award from the NINDS.

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

# A   APPENDIX

## A.1   RAT MEASUREMENTS

To construct the virtual rodent model, we obtained the mass and lengths of the largest body segments that influence the physical properties of the virtual rodent. First, we dissected cadavers of two female Long-Evans rats, and measured the mass of relevant limb segments and organs. Next, we measured the lengths of body segments over the skin of animals anesthetized with 2% v/v isoflurane anesthesia in oxygen. We confirmed that these skin based measurements approximated bone lengths by measuring bone lengths in a third cadaver. The care and experimental manipulation of all animals were reviewed and approved by the appropriate Institutional Animal Care and Use Committee.

| | Animal (#) | | |
| | 63 | 64 | |
| **Body part** | **Mass (g)** | | **Average mass (g)** |
| Hindlimb L | 21 | 26 | 23.5 |
| Hindlimb R | 21 | 26 | 23.5 |
| Tail | 8 | 10 | 9 |
| Forelimb R | 11 | 14 | 12.5 |
| Forelimb L | 12 | 13 | 12.5 |
| Full torso | 176 | 187 | 181.5 |
| Head | 26 | 26 | 26 |
| Upper torso | 78 | 71 | 74.5 |
| Lower torso | 98 | 114 | 106 |
| Torso without organs | 54 | 58 | 56 |
| Intestines and stomach | 22 | 32 | 27 |
| Liver | 26 | 17 | 21.5 |
| Pelvis and kidneys | 74 | 80 | 77 |
| Jaw | 2.43 | 4.70 | 3.57 |
| Skull | 23 | 21 | 22 |
| Tail (base to mid) | 5.92 | 7.20 | 6.56 |
| Tail (mid to tip) | 1.78 | 2.30 | 2.04 |
| Scapula L | 3.19 | 4.70 | 3.94 |
| Humerus L | 6.25 | 4.70 | 5.48 |
| Radius/ulna L | 2.61 | 2.8 | 2.70 |
| Forepaw L | 0.53 | 0.5 | 0.52 |
| Scapula R | 2.23 | 3.9 | 3.07 |
| Humerus R | 6.08 | 6.7 | 6.39 |
| Radius/ulna R | 2.17 | 3.3 | 2.74 |
| Forepaw R | 0.53 | 0.5 | 0.52 |
| Hindpaw L | 1.66 | 1.7 | 1.68 |
| Tibia L | 9 | 9 | 9 |
| Femur L | 13 | 16 | 14.5 |
| Hindpaw R | 1.81 | 1.6 | 1.71 |
| Tibia R | 5 | 6 | 5.5 |
| Femur R | 13 | 18 | 15.5 |
| Total | 281 | 301 | 291 |

Table 1: Before weighing, limb segments were divided at their respective joints. Mass of all segments includes all bones, skin, muscle, fascia and adipose layers. L and R refer to the left and right sides of the animal. Precision of measurements listed without decimal places is $\pm0.5$g

| | | | | Animal (#) | | | | |
|---|---|---|---|---|---|---|---|---|
| | **48** | **62** | **55** | **56** | **64** | **63** | **62*** | **Average $\pm$ std** |
| **Age (days)** | 382 | 82 | 330 | 330 | 83 | 83 | 83 | |
| **Mass (g)** | 325 | 273 | 389 | 348 | 283 | 269 | 273 | 309 $\pm$ 47 |
| **Body part** | | | | **Length (mm)** | | | | |
| Ankle to claw L | 40.2 | 39.5 | 39.7 | 37.8 | 39.9 | 41.5 | 39.8 | 39.8 $\pm$ 1.1 |
| Ankle to toe L | 38.4 | 38.12 | 37.7 | 35.6 | 36.6 | 39.3 | 38 | 37.7 $\pm$ 1.2 |
| Ankle to pad L | 23.4 | 22.2 | 23 | 22.12 | 22.5 | 23.3 | 6.4 | 20.4 $\pm$ 6.2 |
| Ankle to claw R | | 38.2 | 40.4 | 38.3 | 39.3 | 39.6 | 38.3 | 39.0 $\pm$ 0.9 |
| Ankle to toe R | | 37 | 38.7 | 36.3 | 37.7 | 38.6 | 36.2 | 37.4 $\pm$ 1.1 |
| Ankle to pad R | | 22.4 | 23.3 | 21.9 | 21.8 | 23.1 | 24.1 | 22.8 $\pm$ 0.9 |
| Tibia L | 50 | 36.3 | 38.5 | 49.2 | 35.8 | 38.7 | 34.1 | 40.4 $\pm$ 6.5 |
| Femur L | 44.5 | 31.6 | 32.1 | 37.9 | 33.4 | 35.35 | 32.4 | 35.3 $\pm$ 4.6 |
| Tibia R | | 36.7 | 39.1 | 37.9 | 35.1 | 38.4 | 36.18 | 37.2 $\pm$ 1.5 |
| Femur R | | 32.9 | 32.1 | 38.7 | 31.9 | 32.1 | 32.6 | 33.4 $\pm$ 2.6 |
| Pelvis | 25.8 | 32 | 31.7 | 30.2 | 26.7 | 27.2 | | 28.9 $\pm$ 2.7 |
| Wrist to claw L | 15 | 18.8 | 17.6 | 18.6 | 16 | 19.02 | 19.2 | 17.7 $\pm$ 1.6 |
| Wrist to finger L | | 16 | 15.8 | 17.4 | 15.5 | 17.07 | 17.6 | 16.6 $\pm$ 0.9 |
| Wrist to pad L | | 6 | 6.4 | 8.34 | 4.9 | 6.1 | 6.4 | 6.4 $\pm$ 1.1 |
| Wrist to olecranon L | 29.1 | 34 | 32.5 | 31.7 | 33.9 | 32.1 | 29.9 | 31.9 $\pm$ 1.9 |
| Humerus L | 31.9 | 29.52 | 31 | 28.2 | 27 | 31.2 | 25.4 | 29.2 $\pm$ 2.4 |
| Scapula L | 22.7 | 24 | 26.4 | 29.3 | 25.9 | 29.1 | 26.2 | 26.2 $\pm$ 2.4 |
| Wrist to claw R | | 16.8 | 17 | 17.8 | 15.9 | 16.3 | 18.1 | 17.0 $\pm$ 0.8 |
| Wrist to finger R | | 14.1 | 13 | 15.6 | 15.6 | 15.3 | 16.9 | 15.1 $\pm$ 1.4 |
| Wrist to pad R | | 5.6 | 5.8 | 6.55 | 5.2 | 5 | 5.8 | 5.7 $\pm$ 0.5 |
| Wrist to olecranon R | | 30.6 | 33.5 | 31.2 | 30.4 | 31.8 | 29.9 | 31.2 $\pm$ 1.3 |
| Humerus R | | 28.2 | 33.5 | 28.8 | 25 | 28.2 | 25.2 | 28.1 $\pm$ 3.1 |
| Scapula R | | 23.8 | 29.5 | 25.9 | 26.2 | 28.8 | 24.4 | 26.4 $\pm$ 2.3 |
| Headcap width | 39 | | | | | | | 39 |
| Headcap length | 30 | | | | | | | 30 |
| Skull width | 38.8 | 23.35 | 23 | 21.8 | 22.8 | 23.9 | 22.2 | 25.1 $\pm$ 6.1 |
| Skull length | 57 | 51.1 | 61 | 56.48 | 53.16 | 58.13 | 48 | 55.0 $\pm$ 4.5 |
| Skull height | | | | | 21.59 | 21.5 | 21 | 21.4 $\pm$ 0.3 |
| Head to thoracic | | 48.6 | 71.4 | 68.68 | 65 | 60.4 | 71.2 | 64.2 $\pm$ 8.7 |
| Thoracic to sacral | | 73.1 | 73.6 | 62.9 | 65.04 | 64.7 | 68.8 | 68.0 $\pm$ 4.6 |
| Head to sacral | 145 | 126 | 145.5 | 127.05 | 127.2 | 123.7 | 140.9 | 133.6 $\pm$ 9.7 |
| Head width | 53.4 | | | | | | | 53.4 |
| Ear | 18 | 17.55 | 19.3 | 17.9 | 19.2 | 18.8 | | 18.5 $\pm$ 0.7 |
| Eye | 7.2 | 8.25 | 8.6 | 8.8 | 8.2 | 8.3 | | 8.2 $\pm$ 0.6 |

Table 2: Length measurements of limb segments used to construct the virtual rodent model from 7 female Long-Evans rats. Measurements were performed using calipers either over the skin or over dissected bones (*). Thoracic and sacral refer to vertebral segments. L and R refer to the left and right sides of the animal's body.

## A.2 BEHAVIORAL ANALYSIS

We generated features describing the whole-body pose and kinematics of the virtual rodent on fast, intermediate, and slow temporal scales. To describe the whole-body pose, we took the top 15 principal components of the virtual rodent's joint angles and joint positions to yield two 15 dimensional sets of eigenpostures (Stephens et al., 2008). We combined these into a 30 dimensional set of postural features. To describe the animal's whole-body kinematics, we computed the continuous wavelet transform of each eigenposture using a Morlet wavelet spanning 25 scales. For each set of eigenpostures this yielded a 375 dimensional time-frequency representation of the underlying kinematics. We then computed the top 15 principal components of each 375 dimensional time-frequency representation and combined them to yield a 30 dimensional representational description of the animal's behavioral kinematics. To facilitate comparison of kinematics to neural representations on different timescales, we used three sets of wavelet frequencies on 1 to 25 Hz (intermediate), 0.3 to 5 Hz (slow) or 5-25 Hz (fast) timescales. In separate work, we have found that combining postural and kinematic information improves separation of animal behaviors in behavioral embeddings. Therefore, we combined postural and dynamical features, the later on intermediate timescales, to yield a 60 dimensional set of 'behavioral features' that we used to map the animal's behavior using tSNE (Figure 4C) (Berman et al., 2014). tSNEs were made using the Barnes-Hut approximation with a perplexity of 30.

## A.3 POWER SPECTRAL DENSITY OF BEHAVIOR AND NETWORK ACTIVITY

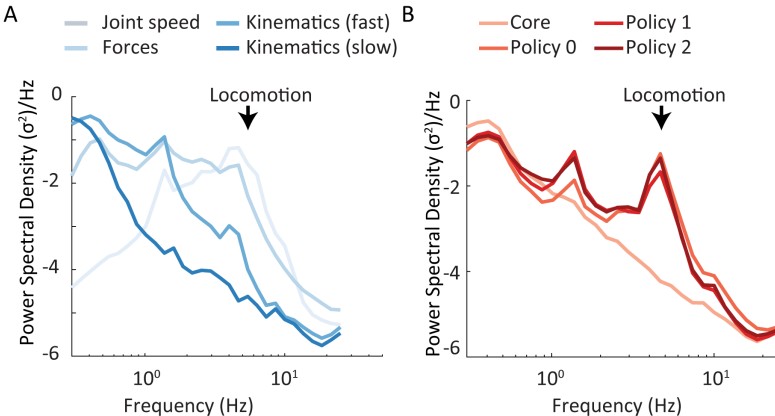

Figure 9: (A) Power spectral density estimates of four different features describing animal behavior, computed by averaging the spectral density of the top ten principal components of each feature, weighted by the variance they explain. (B) Power spectral density estimates of four different network layers, computed by averaging the spectral density of the top ten principal components of each matrix of activations, weighted by the variance they explain. Notice that policy layers have more power in high frequency bands than core layers. Arrows mark peaks in the power spectra corresponding to locomotion. Notably, the 4-5 Hz frequency of galloping in the virtual rat matches that measured in laboratory rats (Heglund & Taylor, 1988). Power spectral density was computed using Welch's method using a 10 s window size and 5 s overlap.

## A.4 Representational Similarity Analysis

We used representational similarity analysis to compare population representations across different network layers and to compute the encoding strength of different features describing animal behavior in the population. Representational similarity analysis has in the past been used to compare neural population responses in tasks where behavioral stimuli are discrete, for instance corpuses of objects or faces (Kriegeskorte et al., 2008; Kriegeskorte & Diedrichsen, 2019). A challenge in scaling such approaches to neural analysis in the context of behavior is that behavior unfolds continuously in time. It is thus *a priori* unclear how to discretize behavior into discrete chunks in which to compare representations.

Formally, we defined eight sets of features $B_{i=1...8}$ describing the behavior of the animal on different timescales. These included features such as joint angles, the angular speed of the joint angles, eigenposture coefficients, and actuator forces that vary on short timescales, as well as behavioral kinematics, which vary on longer timescales and 'behavioral features', which consisted of both kinematics and eigenpostures. Each feature set is a matrix $B_i \in \mathbb{R}^{Mxq_i}$ where $M$ is the number of timepoints in the experiment and $q_i$ is the number of features in the set. We discretized each set $B_i$ using k-means clustering with $k = 50$ to yield a partition of the timepoints in the experiment $P_i$.

Using the discretization defined in $P_i$, we can perform representational similarity analysis to compare the structure of population responses across neural network layers $L_m$ and $L_n$ or between a given network layer and features of the behavior $B_i$. Following notation in (Kornblith et al., 2019) we let $X \in \mathbb{R}^{kxp}$ be a matrix of population responses across $p$ neurons and the $k$ behavioral categories in $P_i$. We let $Y \in \mathbb{R}^{kxq}$ be either the matrix of population responses from $q$ neurons in a distinct network layer, or a set of $q$ features describing the behavior of the animal in the feature set $B_i$.

After computing the response matricies in a given behavioral partition, we compared the representational structure of the matricies $XX^T$ and $YY^T$. To do so, we compute the similarity between these matricies using the linear Centered Kernel Alignment index, which is invariant under orthonormal rotations of the population activity. Following (Kornblith et al., 2019), the CKA coeffient is:

$$CKA(XX^T, YY^T) = \frac{\|XY^T\|_F}{\|XX^T\|_F \|YY^T\|_F} \tag{1}$$

Where $\| \cdot \|_F$ is the Frobenius norm. For centered $X$ and $Y$, the numerator is equivalent to the dot-product between the vectorized responses $\|XY^T\|_F = \langle \text{vec}(XX^T), \text{vec}(YY^T) \rangle$.

For a given network layer $L_m$, and a behavioral partition $P_i$, we can denote $XX^T = D_{P_i}^{L_m} = D_i^m$. Similarly, for a given feature set $B_i$, let $D_{P_i}^{B_i} = D_i^i$. Thus we are interested in characterizing both

$$CKA(D_i^m, D_i^n) \tag{2}$$

and

$$CKA\left(D_i^m, D_i^i\right). \tag{3}$$

The former equation describes the similarity across two layers of the network, and the later describes the similarity of the network activity to a set of behavioral descriptors.

An additional challenge comes when restricting this analysis to comparing the neural representations of behavioral across different tasks $T_a, T_b$, where not all behaviors are necessarily used in each task. To make such a comparison, we denote $B_i(T_a)$ to be the set of behavioral clusters observed in task $T_a$, and $B_i^{T_a T_b} = B_i(T_a) \cap B_i(T_a)$ to be the set of behaviors used in each of the two tasks. We can then define a restricted partition of timepoints for each task $P_i^{T_a, T_b}$ or $P_i^{T_b, T_a}$ that includes only these behaviors, and compute the representational similarity between the same layer across tasks:

$$CKA\left(D_{i,T_a}^m, D_{i,T_b}^m\right). \tag{4}$$

We have presented a means of performing representational similarity analysis across continuous time domains, where the natural units of discretization are unclear and likely manifold. While we focused on analyzing responses on the population level, it is likely that different subspaces of the population may encode information about distinct behavioral features at different timescales, which is still an emerging domain in representational similarity analysis techniques.

A.5    NEURAL POPULATION ACTIVITY ACROSS TASKS DURING RUNNING

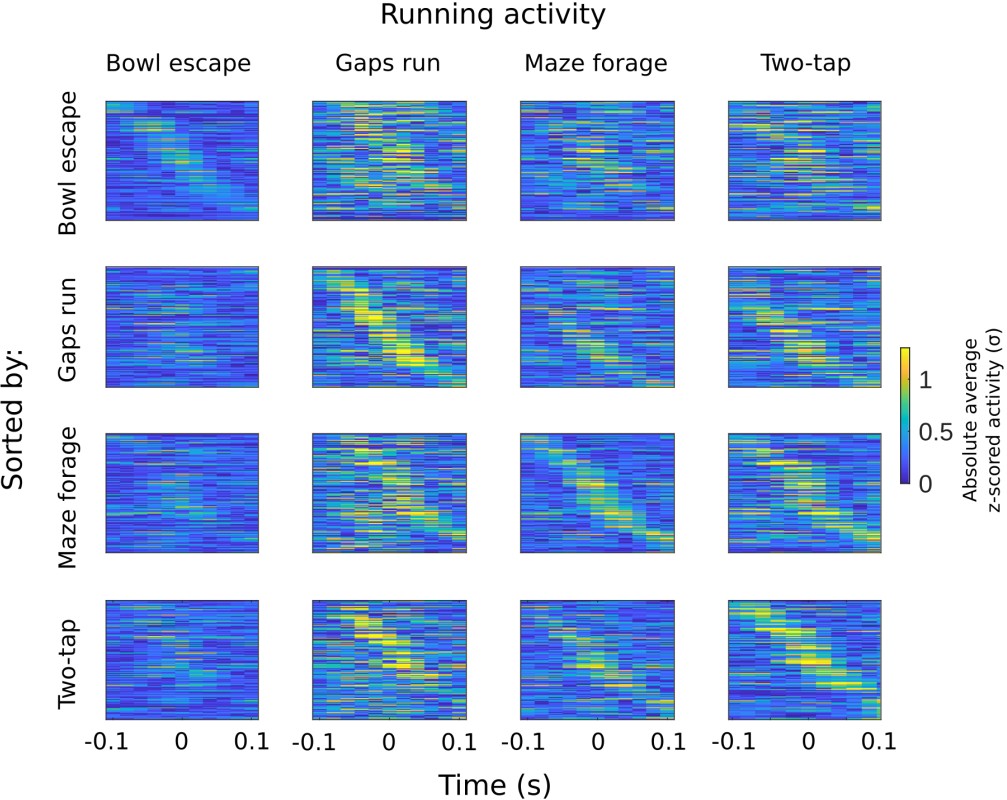

Figure 10: Average activity in the final policy layer (policy 2) during running cycles across different tasks. In each heatmap, rows correspond to the absolute averaged z-scored activity for individual neurons, while columns denote time relative to the mid stance of the running phase. Across heatmaps, neurons are sorted by the time of peak activity in the tasks denoted on the left, such that each column of heatmaps contains the same average activity information with rearranged rows. Aligned running bouts were acquired by manually segmenting the the principal component space of policy 2 activity to find instances of mid-stance running and analyzing the surrounding 200 ms.

## A.6  STEREOTYPED BEHAVIOR INITIATION AND NEURAL VARIABILITY

During the execution of stereotyped behaviors, neural variability was reduced (Figure 11). Recall that in our setting, neurons have no intrinsic noise, but inherit motor noise through observations of the state (i.e. via sensory reafference). This effect loosely resembles, and perhaps informs one line of interpretation of the widely reported phenomenon of neural variability reducing with stimulus or task onset (Churchland et al., 2010). Our reproduction of this effect, which simply emerges from training, suggests that variance modulation may partly arise from moments in a task that benefit from increased behavioral precision (Renart & Machens, 2014).

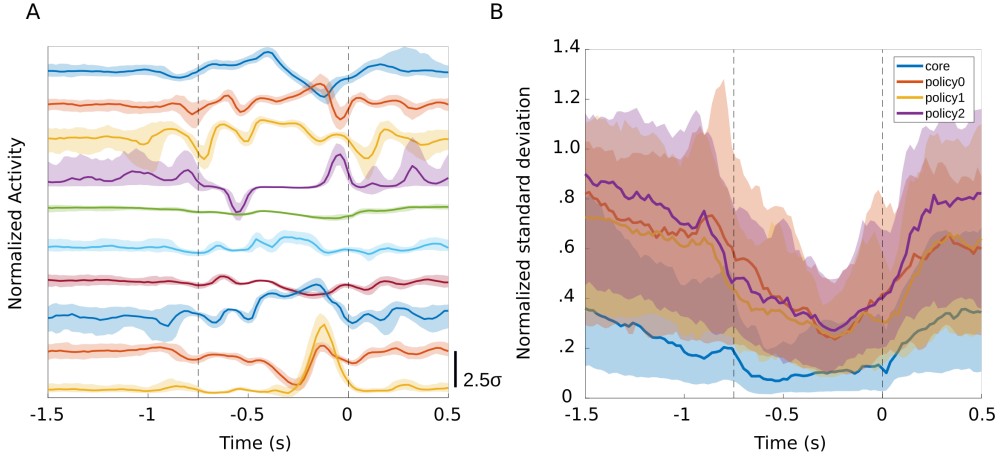

Figure 11: Quantification of neural variability in inter-tap interval of two-tap task relative to the second tap. (A) Example normalized activity traces of ten randomly selected neurons in the final policy layer. Lines indicate mean normalized activity whiles shaded regions range from the 20th percentile to the 80th percentile. Dashed lines indicate the times of first and second taps. (B) Standard deviation of normalized activity across all neurons in the final policy layer as a function of time relative to the second tap. Lines indicate the mean standard deviation while shaded regions range from the 20th percentile to the 80th percentile. Observe that variability is reduced during the two-tap interval.

## A.7  NEURAL DYNAMICS VISUALIZED DURING TASK BEHAVIOR

For completeness, we provide links to videos of a few variants of neural dynamics for each task.

| Network | Visualization | Task (link) |
|---|---|---|
| 1-layer policy | PCA | gaps |
| | PCA | forage |
| | PCA | escape |
| | PCA | two-tap |
| 3-layer policy | PCA | gaps |
| | PCA | forage |
| | PCA | escape |
| | PCA | two-tap |
| 3-layer policy | jPCA | gaps |
| | jPCA | forage |
| | jPCA | escape |
| | jPCA | two-tap |

Table 3: Links to representative visualizations of neural dynamics and behavior

## A.8   PERTURBATION RESULTS

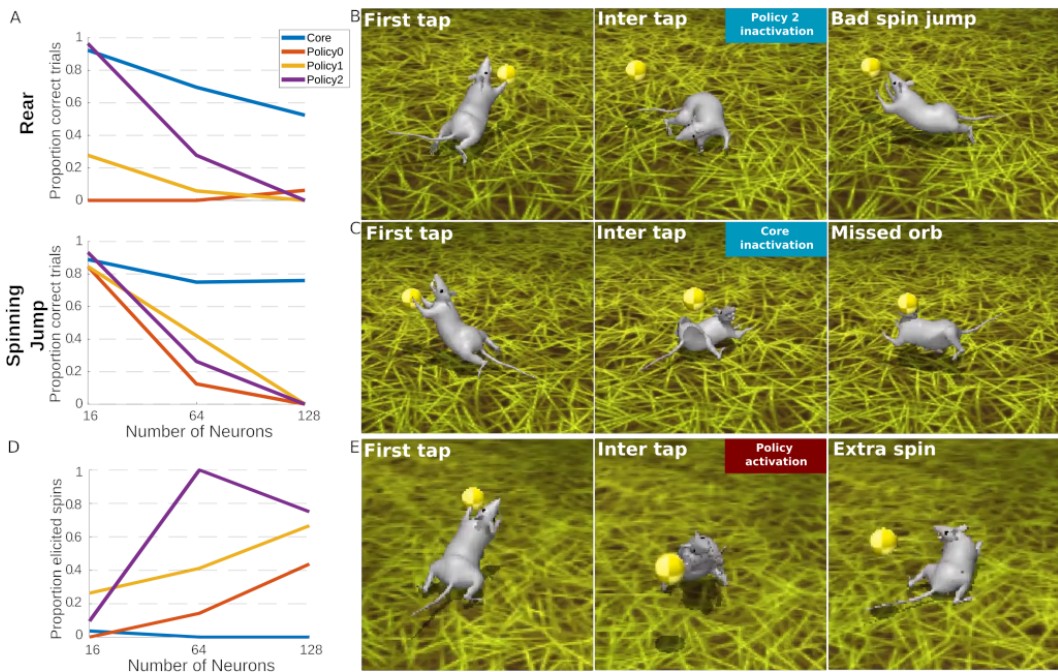

Figure 12: Causal manipulations reveal distinct roles for core and policy layers in the production of behavior. (A) Two-tap accuracy during the inactivation of units modulated by idiosyncratic behaviors within the two-tap sequence. Core inactivation has a weaker negative effect on trial success than policy inactivation for several levels of inactivation. (B) Representative example of a failed trial during inactivation of the final policy layer in a model that performs a spinning jump during the two-tap sequence. The model is incapable of producing the spinning jump behavior while inactivated. (C) Representative example of a failed trial during core inactivation in a model that performs a spinning jump during the two-tap sequence. The model is still able to perform the spinning jump behavior, but misses the orb. (D) Proportion of attempts at stimulation that successfully elicited spin behavior during the two-tap task. The efficacy of this activation was more reliable in layers closer to the motor output. (E) Representative example of a single trial in which an extra spin occurs after policy 2 activation.

