# OpenReview forum: "Deep neuroethology of a virtual rodent"
_ICLR.cc/2020/Conference — Accept (Spotlight)_

### Official Review · AnonReviewer1 · 2019-10-21
**Official Blind Review #1**

**Rating:** 8

**Review:**

The authors use a virtual rodent to study flexible behavior, a key concept at the intersection of AI and neuroscience. “Flexibility” is a common buzzword in this field used to describe what it is that mammals do better than deep nets, even as deep nets can outperform them on any one task in isolation with sufficient training. As the authors note, previous work addresses “cognitive flexibility” (Yang et al. 2019) in isolation without motor commands, or studies motor dynamics of real mammals but without any task flexibility—occluding the actual advantages that mammals have over machines. However, this work goes the full mile in requiring the agent to implement flexible cognition using a complex body in a physical environment with egocentric stimuli. From a strictly AI point of view, I could imagine arguments that this is not sufficiently “novel,” in that a lot of deep RL researchers have already trained agents with complex joint structures to succeed in such virtual worlds. But the real substance of the paper is the application of data analysis techniques imported directly from neuroscience. It is important to suss out what features of neural dynamics are peculiarities of biological computation and which are natural consequences of the tasks being performed. I find the paper to be overall well motivated.

The analyses were well ordered, starting with a high-level characterization of behavior that showed a hierarchy of features, where “low-level” or “fast” features are mixed across tasks while “high-level” or “slow” features are more task specific. They then moved on to find “neural correlates” at the behavior and task levels, driving home their primary point that core layers encode high-level variables in a way not shared across tasks while policy layers implement reusable core motor functions that are used in multiple contexts. This observation coincides with the measurable frequency content of core and policy activity, which is demonstrably faster in policy layers. Finally, the authors show a visualization of the dynamical systems that underlie these computations, with videos that very clearly illustrate how core and policy dynamics dissociate to specialize in cognition vs. motor generation. This part is where they really leverage the artificial setting, where the exact dynamics are known and do not need to be inferred from noisy neural data.

While the end result of the analyses led to convincing visuals that made the intended point, the details of the analysis themselves were hard to follow and evaluate for rigor. An appendix explaining with notation how the behavioral “features” were computed, for example, would have been helpful. I acknowledge that it is a feature of the paper that so many different analysis techniques from neuroscience—dynamical systems to representation analysis to lesion studies—were applied, and the sheer breadth of the analysis was impressive, if difficult to evaluate.

Although I recognize that to do so would be difficult, I think what would add a lot to this paper would be side-by-side comparisons of real neural data with “neural” data from the artificial rodent. Perhaps the behavior would have to be modified so that some tasks match specific neuroscience paradigms, or the sample size for analysis limited to what is realistic for experimental data. It would be neat to see if, say, preparatory activity signals could emerge from training such an artificial rodent to do many tasks, including a head-fixed timed lever pull or some other classic paradigm. Maybe future work can draw a more exact analogy between policy/core layers and specific brain regions.

As to topic fit, this is niche in the context of ICLR; however, showing how state-of-the-art deep RL techniques can be applied to neuroscientific questions may be of interest to machine learning researchers even if they are not also neuroscientists.

Overall, I thoroughly enjoyed this paper and look forward to the new kinds of research that sprout out of it!

**Experience Assessment:**

I have read many papers in this area.

**Review Assessment: Checking Correctness Of Derivations And Theory:**

N/A

**Review Assessment: Checking Correctness Of Experiments:**

N/A

**Review Assessment: Thoroughness In Paper Reading:**

N/A

---

> ### Author Response · Authors · 2019-11-11
> **Specific responses to reviewer 1**
>
> We thank the reviewer for their comments, and we find the perspective articulated in this review very much in line with what we have been setting out to do!  For us, it is satisfying that we now find ourselves in a position to build a somewhat plausible, albeit simplified, virtual rodent.  It was nontrivial to produce this model and multi-task policy, but we share the reviewers enthusiasm for the analysis element of the work, as well as the broader aim of using this class of approach as a model in neuroscience.
>
> We accept that the details of parts of the analyses were not well optimized for presentation in the original submission and in the revised manuscript have clarified the Analysis and Discussion sections to more directly highlight the major findings. We have also, as suggested, expanded on the details of the methods used in supplementary sections in the revised draft (See Appendix A2, A3).  We hope the additional sections are completely satisfying, but please follow-up if anything remains inadequately addressed.
>
> We absolutely share the reviewer’s interest in comparisons with real data and real brain anatomy.  This has been the driving force behind our study, and remains a clear target for future work, about which we are enthusiastic. However, it is beyond the scope of the current submission.

---

### Official Review · AnonReviewer2 · 2019-10-22
**Official Blind Review #2**

**Rating:** 6

**Review:**


=============================== Update after rebuttal ======================================================

I thank the authors for their rebuttal and the revisions. I'm not entirely satisfied with the authors' response to my request for more architectural exploration, but I understand that there wasn't really enough time for this during the relatively short rebuttal period. I think the results in the paper are still valuable enough to merit publication, so I'm happy to increase my score and recommend accepting the paper. However, I still encourage the authors to take the issue of arhcitectural plausibility more seriously in the future, especially if the intended audience for this line of work is experimental neuroscientists. Currently, the architecture choice seems to be dictated primarily by trainability considerations (more specifically, trainability by current deep learning methods).

========================================================================================================

This paper introduces a virtual rodent model with a complex set of actuators and visual and proprioceptive inputs. The model is simultaneously trained on four different tasks and the trained model is analyzed using various dimensionality reduction and visualization methods. The effort put into training and analyzing the rodent model is quite impressive. Moreover, the tools that the authors will make public can be useful for other researchers in this field.

However, my main concern about the paper is that given the architecture choice made in the paper, most of the main results do not seem very surprising. On the other hand, the architecture choice itself is not motivated well enough. The differences between the dynamic and representational properties of the core and policy networks entirely depend on the fact that the core network is trained separately from the policy network. Why was this particular choice made? In a more realistic scenario (for example, in the actual brain of a rodent), everything will presumably be connected to most everything else to a certain degree, with no sharp separation between policy and core modules and the error signals flowing more broadly across the entire network. It seems to me that in such a scenario, there wouldn't be such a big difference between the dynamic and representational properties of different modules in the network. So, I am wondering if it is possible to train more models with alternative architectures (with presumably more realistic properties) and compare the results with the current results.

**Experience Assessment:**

I have published in this field for several years.

**Review Assessment: Checking Correctness Of Derivations And Theory:**

N/A

**Review Assessment: Checking Correctness Of Experiments:**

I assessed the sensibility of the experiments.

**Review Assessment: Thoroughness In Paper Reading:**

I read the paper at least twice and used my best judgement in assessing the paper.

---

> ### Author Response · Authors · 2019-11-11
> **Specific responses to reviewer 2**
>
> We thank the reviewer for acknowledging both the effort and for their assessment that the tools developed in this work will be useful for others.  We do hope that others will expand on what we’ve done using the rodent body (which, as acknowledged by the reviewer, will be released).
>
> The main concerns of the reviewer have to do with the choice of architecture and that our choice makes the results obtained in the analysis unsurprising.  We’ve attempted to address the concern about which elements of our results were not predictable, even if they make sense in hindsight, in the general response to reviewers.  Below, we address the issues concerning our specific choice of architecture.
>
> Is our architecture realistic?
> We share the reviewer’s enthusiasm for exploring different, and more realistic architectures.  We view it as part of a longer enterprise to reverse engineer the functional architectures present in real nervous systems.  In some sense, our goal here is to start on a path of iterative refinement of synthetic architectures.  That being said, contrary to the reviewer, we do not believe the most realistic architectural choice is a fully connected and entirely end-to-end differentiable architecture.  Real brains have different, functionally localized subsystems, that are likely optimizing different objectives.  We defend as reasonable the choice to have a single architecture which must both predict value and produce actions, and we also believe it is reasonable for there to be some architectural segregation of these pathways.  For example, in the basal ganglia there is thought to be some segregation between the prediction of value and action selection (Joela et al 2002).
>
> Somewhat separately, it is relatively common in DeepRL for parts of a network to be shared by the value function and policy, so our architectural choice is not really unconventional from that standpoint either.
>
> Why not try more architectures?
> In order to be systematic, we sought to train all of our models comparably and conduct thorough analyses.  Accordingly, we had to judiciously restrict the diversity of architectures we could assess.  As noted in the paper, we focused on two architectural choices: with and without skip connections in the policy layers and different numbers of policy layers.  This gave us the ability to test how increasing or decreasing the network capacity affected the representations networks learned.  Nevertheless, we also agree that in future work, it will be informative to explore an even wider set of architectures.  Future exploration of architectures may be motivated either to test specific hypotheses about biology or enable functionality/capabilities required for additional tasks.
>
> Reference:
> “Actor–critic models of the basal ganglia: new anatomical and computational perspectives”. Daphna Joela, Yael Niv, Eytan Ruppin (2002)

---

### Official Review · AnonReviewer3 · 2019-10-28
**Official Blind Review #3**

**Rating:** 6

**Review:**

This is a fascinating paper that uses methods from computational neuroscience to characterize a neural network that controls a virtual rat (or, well, something ratlike). I really like the idea of trying to fuse neuroethology and animal behavior research in general with deep learning methods. To me, it seems like there was plenty of activity in this field around 20 years ago (with e.g. the robot models of animal behavior of Ijspeert and others, and the evolutionary robotics approach to study the evolution of behaviors) but that this research line has not merged with (alternatively informed, or learned from) modern deep learning. So I welcome this direction of research. This being said, it's not my field of research, so I'm unable to comment on several of the specifics here.

The learning of a network that can perform these four independent tasks is quite impressive in its own right. I would like the paper to comment on how hard or easy this was, if you attempted to learn other behaviors but failed, etc.

The actual analysis is a bit of a letdown - not because it seems to be wrong or incompetently done, rather the opposite - but in that there is so little to learn from it. Simply put, I do not understand anything more about networks that control simulated robots to perform multiple tasks (or about real rodents) after reading the paper. What does this mean? Is this an indictment of current neuroscience methods, that even when you have unambiguous non-noisy access to all of the state of the network, you cannot really find out much? If so, you should discuss this. If not, you should explain what's going on. At least from the perspective of an AI researcher, there just isn't enough understanding there.

However, I don't think negative results (which this in a way is) should discourage from publication. I applaud the intent of the paper, the competence with which it was executed, and the learning of the network. So I want to see this published. But please remark on the points above.

**Experience Assessment:**

I have read many papers in this area.

**Review Assessment: Checking Correctness Of Derivations And Theory:**

N/A

**Review Assessment: Checking Correctness Of Experiments:**

I assessed the sensibility of the experiments.

**Review Assessment: Thoroughness In Paper Reading:**

I read the paper at least twice and used my best judgement in assessing the paper.

---

> ### Author Response · Authors · 2019-11-11
> **Specific responses to reviewer 3**
>
> We appreciate the reviewer’s thoughtfulness concerning the context in which this work was conducted as well as the assessment that it is a useful, and presently underexplored, line of research.
>
> Concerning learning of a multi-task policy, we accept that it may not be obvious to a reader how easy it is to obtain this policy.  For the present work, these were indeed the four tasks we aimed to solve, and given some experience with these techniques, this was not particularly finicky to get working.  However, as mentioned in the original submission, training interleaved on all four tasks worked but was not 100% reliable across seeds. As a consequence, we used kickstarting for the bowl-escape task, which made the training of four tasks entirely reliable across seeds and architectures (this is stated in the paper). Speaking to another point raised elsewhere concerning training other architectures, we have not been entirely systematic, but it is definitely the case that certain changes to the architecture make it impossible to train a single policy to solve all four tasks.  It is not straightforward for us to survey all ways of breaking the training procedure, but what we have presented is the result of consideration of some alternatives.
>
> We hope that our general reply to all reviewers highlights for you what we saw as the main messages of our analyses.

---

### Author Response · Authors · 2019-11-11
**General responses to all reviewers**

We thank all of the reviewers for their thoughtful assessments and feedback.  Overall, we found the reviews constructive, and generally aligned with our own interpretations of our work.  We directly responded to each reviewer and have updated the draft to clarify our analyses, results, and interpretations.

One theme that came up in all reviews is related to how “expected” or “surprising” the results of the analyses are. In the revised manuscript we have amended the figures as well as the Analysis and Discussion sections to more directly emphasize what we feel are the messages to take away from the analyses. We designed our analyses of the virtual rodent to identify the types of neural representations and dynamics that can be used by real or artificial systems to generate multiple behaviors using a shared body.  We also aimed for this analyses to enable us to relate the structure that we identified within the artificial neural activity to previous studies from neuroscience.  Our stance on this point is that the set of analysis methods that we selected are indeed performing somewhat successfully.  While in hindsight, none of the results seem alarming, and a subset might be predictable in light of what is known with full knowledge of the architecture, objective functions, and tasks the virtual animal was trained on, we do believe that some analyses revealed properties of the neural activity that were not obvious a priori, and these findings relate to existing ideas from neuroscience.

Some key results of the analyses:
- Even in the policy layers there was not very strong coding of low-level motor features such as actuator forces or joint-angle positions and velocities.
- Policy activity is more similar across tasks than is core activity.  The core activity is very different across tasks.  And in particular, network capacity seems to affect the amount of similarity of behavior-specific neural activity across tasks.
- Behaviors on different timescales are associated with rotations in latent spaces that follow stable orbits. And these orbits relate to emergent sequential activity (e.g., fig 7).

We now more clearly highlight these points as we proceed through the analyses.

Relatedly, we find that some features of real neural systems naturally emerge in our system, without being explicitly encouraged.  To name a few specific findings from neuroscience, our observation of emergent sequential and sparse neural activity in the production of stereotyped behaviors parallels findings across several model systems in neuroscience including rodents and songbirds. Similarly, the observation of latent rotational dynamics associated with the production of different behaviors has been previously described in studies of monkey motor neuroscience.  We’ve incorporated and contextualized these connections with the neuroscience literature in the main text Discussion section (with references).   We are optimistic about future efforts which will aim for direct comparisons with behaving animals, but this is beyond the scope of the present submission.

Finally, we wish to emphasize that in neuroscience, methods are rarely tested in ground-truth settings, with full observation of inputs and outputs, a known architecture, as well as knowledge of what the agent is optimized to do.  While it is hard to defend completely against claims that all results were expected (especially in light of the architecture and training details), the methods we have employed did not have access to this information. In that sense, our setting enables a validation of analysis methods, enabling careful assessment of consistency between our intuitions of how the architecture “should” work and what the analyses reveal.  We’ve made this additional motivation clearer in our revision.

---

### Decision · Program_Chairs · 2019-12-19

**Decision:**

Accept (Spotlight)

**Comment:**

This paper is somewhat unorthodox in what it sets out to do: use neuroscience methods to understand a trained deep network controlling an embodied agent. This is exciting, but the actual training of the virtual rodent and the performance it exhibits is also impressive in its own right. All reviewers liked the papers. The question that recurred among all reviewers was what was actually learned in this analysis. The authors responded to this convincingly by listing a number of interesting findings.

I think this paper represents an interesting new direction that many will be interested in.